# Developing Future Clinical Pharmacy Leaders in the Interprofessional Care of Children with Special Health Care Needs and Medical Complexity (CSHCN-CMC) in a Pediatric Pulmonary Center

**DOI:** 10.3390/children6120135

**Published:** 2019-12-09

**Authors:** Catherine B. Hobart, Cori L. Daines, Hanna Phan

**Affiliations:** 1Department of Pharmacy, Banner University Medical Center Tucson, Tucson, AZ 85719, USA; Catherine.hobart2@bannerhealth.com; 2Department of Pharmacy Practice and Science, College of Pharmacy, The University of Arizona, Tucson, AZ 85721, USA; 3Section of Pediatric Pulmonary and Sleep Medicine, Department of Pediatrics, College of Medicine, The University of Arizona, Tucson, AZ 85724, USA; cdaines@email.arizona.edu

**Keywords:** children with special health care needs and medical complexity (CSHCN-CMC), pediatric ambulatory care pharmacist, collaborative practice, cystic fibrosis, pulmonary disease, interprofessional, maternal child health workforce development

## Abstract

The health care needs of children with special health care needs and medical complexity (CSHCN-CMC) are multifaceted and often require the expertise of various disciplines. The medication-related needs of this population can be further complicated with off-label medication use, polypharmacy, and vulnerability to medication errors. Although clinical pharmacists are increasingly becoming a common part of inpatient, pediatric interprofessional patient care teams, their presence remains lacking in the outpatient or ambulatory care realm. Pediatric clinical pharmacists in the ambulatory care setting have the potential to help optimize medication use and safety through collaborative efforts as part of the interprofessional team. Since the late 1960s, Pediatric Pulmonary Centers (PPCs) provide training programs designed to develop interprofessional leaders who will improve the health status of CSHCN-CMC, specifically those with chronic respiratory and sleep-related conditions. The addition of pharmacists not only provides a more comprehensive care model for CSHCN-CMC, it creates an avenue to encourage the career paths of pediatric pharmacists in the ambulatory care setting. Here, we describe the addition of clinical pharmacy as part of an interprofessional patient care team and the development and implementation of a maternal child health (MCH) pharmacy discipline training model designed to mentor future pharmacist leaders in the care of CSHCN-CMC.

## 1. Introduction

Interprofessional patient care requires the contribution of various skills, knowledge, and experiences from different health care staff members to come together, providing care to a patient. The prevalence of interprofessional health care has increased to eliminate barriers as well as optimize patient/family-centered care. Common goals established by the interprofessional team are achieved through interdependent collaboration, open communication, and shared decision making through establishing common health goals, assessments, and planning when caring for patients with complex needs [1]. 

Children and youth with special health care needs (CSHCN) and children with medical complexity (CMC) have been defined as those that “have or are at increased risk for chronic physical, developmental, behavioral or emotional conditions and who also require health and related services of a type or amount beyond that required by children generally” [2]. Nearly 15 million (19.8%) children have special health care needs, including complex service needs and/or multiple prescription medications to manage health condition(s) [3]. Health care utilization of children with special health care needs and medical complexity (CSHCN-CMC) is greater than their counterparts including higher care coordination and mental/educational health care services. Health outcomes are also worse, including overall health, oral health, and need for emergency care [4,5]. Given the complex nature of the care of CSHCN, interprofessional patient care teams are often used such as that seen in palliative care [6]. The health care discipline composition of said patient care teams, however, is not well defined nor universal. Although clinical pharmacists are increasingly becoming a common part of inpatient interprofessional patient care teams, their presence remains lacking in the outpatient or ambulatory care realm, especially in the care of children. The lack of widespread integration of clinical pharmacists in the pediatric ambulatory care setting may be due to various factors including limited data describing their role and impact, current limitations of reimbursement, and ongoing challenge of support to create and sustain pediatric clinical pharmacy services in this setting. Pediatric clinical pharmacists in the ambulatory care setting offer unique expertise to help optimize medication efficacy and safety through collaborative efforts as part of the interprofessional team. Recognizing the importance of pharmacy inclusion and active participation on the interdisciplinary team in the inpatient and outpatient setting, it is critical to have training programs and educational opportunities to help shape the future leaders of pharmacy practice.

## 2. Pediatric Pulmonary Centers (PPCs): Contributing to Interprofessional, Maternal Child Health (MCH) Workforce Development 

The development of future maternal child health (MCH) professionals is recognized through various workforce development initiatives, including those supported by the Health Resources and Services Administration (HRSA). The HRSA serves as the primary U.S. agency responsible for improving health care for people that are vulnerable. Vulnerability may result from geographic isolation, economic hardships, or medically vulnerable people. The HRSA strives, “to improve health outcomes and address health disparities through access to quality services, a skilled health workforce, and innovative, high-value programs” [7,8]. To achieve the goals set forth by the HRSA, this governing agency contributes to the training of health professionals to distribute providers in the areas in the most need. Functioning under the HRSA, the Maternal and Child Health Bureau (MCHB) envisions, “an America where all children and families are healthy and thriving and have a fair shot at reaching their fullest potential” [8]. The MCHB implements various programs focusing on specialty areas of maternal/women’s health, adolescent/young adult health, perinatal/infant health, children with special health care needs, and child health to accomplish the organization’s intentions. Pediatric Pulmonary Centers (PPCs) are funded projects under the Division of Maternal Child Health, Workforce Development, which prepares future health care professionals to develop or improve community-based, family-centered health care for children with chronic respiratory diseases such as cystic fibrosis and asthma (Figure 1) [9,10,11,12,13].

PPCs emphasize interprofessional training, system integration, and research, ultimately, striving to develop future health care leaders with the ability to cultivate and improve community-based, family-centered care for children with chronic respiratory conditions. PPCs work to help eliminate barriers to health, improve health infrastructure, and address health disparities to promote family-centered, interprofessional care. There are currently six active PPC project locations throughout the U.S. (Alabama, Arizona, Florida, New Mexico, Washington, and Wisconsin). In supporting the broad reach of PPCs throughout the country, all PPCs collaborate with various state Title V (MCH) agencies or other MCH-related programs. Title V is a federal program that partners with state MCH and CSHCN programs to improve health for all mothers and children [12]. Collaboration efforts encompass service, training, continuing education, technical assistance, product development and research [13]. PPCs provide training at the graduate and post-graduate levels in the five core disciplines: medicine, nursing, social work, nutrition, and family leadership. Additional disciplines, such as pharmacy, have been added at some PPC training sites [9,10,11,12,13]. 

## 3. The University of Arizona Pediatric Pulmonary Center Interprofessional Patient Care Model 

In July 2000, The University of Arizona (UA) PPC (Tucson, Arizona) was established, initially including medicine, nursing, social work, and nutrition as the core disciplines. Each discipline has a designated faculty mentor overseeing the trainees’ curriculum and clinical training. PPC faculty mentors are clinicians who are part of our patient care team and appointed faculty in their respective disciplines’ programs at The University of Arizona [14]. For medicine, trainees include pediatric pulmonary fellows and pediatric medicine residents. Nursing, social work, and nutrition trainees are often current graduate students (e.g., Master of Social Work, Master of Science in Nutrition, Master or Doctor of Philosophy in Nursing, Doctor of Nurse Practitioner). The Family Leadership discipline was added to the program and thereby highlighting the importance of family/patient perspective to the care of CSHCN-CMC. Family leaders are representatives that have a direct family member with a complex pulmonary need. These family leaders are impacted by the care provided by the interdisciplinary team and participate in the decision-making process. Including family leaders in the PPC helps facilitate co-production of care in the clinical setting and provides insight to ensure patient and family perspectives are heard and incorporated into care. To continue the growth and development of the traineeship, and in recognizing the importance of safe and effective medication use in the care of CSHCN-CMC, the UA PPC expanded their clinical care and training program to include the pharmacy discipline in 2009. 

The decision to add the pharmacy discipline to the PPC was based on the experienced benefit of a clinical pharmacy specialist as part of the pediatric pulmonary consult inpatient service, by the PPC Center Director, a physician, as well as the attending physician team. The attending physicians recognized the clinical pharmacy specialist’s expertise to optimize both acute and chronic therapy and acknowledged the need for this approach to care in the outpatient setting. As a result, the pharmacy discipline was added to the PPC and its outpatient clinics. In this role, the clinical pharmacy specialist aids in medication management, education, monitoring for indications to initiate or modify medications, as well as safety and efficacy. The clinical pharmacy specialist contributes to quality improvement initiatives and continually works to eliminate patient barriers to accessing a clinical pharmacist and medications. Since the commencement of the program in 2000, the UA PPC has trained over 50 trainees, of which 14 were pharmacy student or resident trainees.

At the UA PPC, various disciplines are part of the patient care model for the pediatric cystic fibrosis (CF) clinic (Figure 2). Our CF care team’s purpose emphasizes the partnership between the care team and patient/family—“We work together and learn with our patients and families to produce the best possible care for people living with CF.” In practice, our CF care team practices “coproduction of care,” where the patient and family are partners in developing and implementing care plans. Both patient and family engagement are highly valued as we believe this is vital towards positive care outcomes. In the UA PPC patient care model, each discipline contributes to the total body of care by contributing their expertise and collaborating with one another to ensure the most comprehensive care. At each CF clinic visit, patients/families have the opportunity to meet with each discipline one-on-one to discuss their questions and concerns. Typically, patients may expect providers in the following order: respiratory therapy, clinical pharmacy, nutrition, social work, psychologist, physician, nurse coordinator, although there may be some variation based on the patient’s needs. From the discussion with the providers, patients/families and the care team collaboratively develop a care plan to help meet the patient and family’s overall needs and health goals. All clinicians from various disciplines and trainees share a common workspace during clinics to encourage interprofessional collaboration of care. Workflow processes were developed and are continually refined to ensure effective yet time efficient clinic visits for patients and families. Recent clinic time evaluations revealed clinic appointments typically last up to two hours, which varied depending on patient’s needs and case complexity. To further improve this process, continual quality improvement initiatives for our CF care center were developed and implemented by the collective group, including additional patient and family advisory council members [15]. The addition of the pre-visit planning phone calls conducted prior to clinic visits is a recently added quality improvement initiative that involves a certified pharmacy technician completing medication reconciliations prior to clinic arrival and identify specific patient needs to address during the scheduled visit (see Section 3.2 Growth of the Pharmacy Care Model for CSHCN-CMC).

### 3.1. The Interprofessional Patient Care Model and the Addition of a Clinical Pharmacy Specialist 

Cystic fibrosis (CF) is a multi-organ genetic disease resulting in time-consuming treatment regimens, frequent hospitalizations, and disease progression. Due to high symptom burden and the increasing presence of comorbidities, the treatment for CF is often elaborate and may require a multitude of medications, physical treatments, and routine visits with CF specialists contributing to treatment complexity. Proper treatment is essential to the management of CF. However, patients with CF may face challenges, including barriers of knowledge and access, that prohibit adherence to such complex treatment regimens which negatively impacts health outcomes [16]. In 2009, the UA PPC added a pharmacy discipline faculty member to their center, who also serves as faculty for the College of Pharmacy and College of Medicine. This individual developed various clinical pharmacy services as part of the interprofessional pediatric CF and severe asthma clinics. 

The one designated clinical pharmacy specialist works in tandem with the attending physicians in managing patients’ plan of care in a collaborative fashion. A collaborative practice model serves as an agreement between the attending physician(s) and clinical pharmacy specialist. This allows for the clinical pharmacy specialist to be responsible for complex medication management, including assessment of current therapy for clinical appropriateness, adherence to clinical care guidelines and as appropriate, initiating, monitoring, modifying, and discontinuing medications, drug interaction management, adherence assessment and intervention, as well as patient/family education in the pediatric CF and severe asthma clinics. Similar to other states that permit collaborative practice models, as per the Arizona State Board of Pharmacy law, pharmacists may initiate, monitor, and modify drug therapy when following written drug therapy management protocols agreed upon with the diagnosing provider [17]. At the UA PPC, the clinical pharmacy specialist manages medications such as cystic fibrosis transmembrane conductance regulator modulating agents, pancreatic enzymes, anti-inflammatory agents, inhaled medications, antibiotics and associated drug monitoring such as therapeutic drug monitoring or pharmacokinetic monitoring. Services span the outpatient setting, inpatient setting, and transitions of care. 

The Clinical Pharmacy Specialist typically sees up to 16 patients during a full CF clinic day with the CF interprofessional care team. These encounters may be classified as routine clinic visits or follow up sick visits. In addition to chronic medication management, the clinical pharmacy specialist is a core team member involved in outpatient acute exacerbation treatment including selection and dosing of oral and/or intravenous home antibiotic therapy as well as inpatient treatment of acute pulmonary exacerbations. In evaluating this, we conducted a 6-month retrospective chart review of the adult and pediatric patients in the CF Center, evaluating patient outcomes before and after the added involvement of a clinical pharmacy specialist in the admission processes for acute pulmonary exacerbations, with data analyzed using descriptive statistics as well as generalized linear and logistic regression analyses with alpha a-priori set at 0.05. The post-period showed improved patient outcomes including decreased length of stay (median +/− interquartile range (IQR)) from 19.4 +/−15.9 days prior to pharmacist involvement to 14.1 +/− 9.3 days (*p* = 0.003), with clinical pharmacy specialist involvement and higher baseline lung function identified as significant factors associated with lower length of stay in regression analyses. The post-period also demonstrated improved medication safety by employing appropriate monitoring strategies within 24 h of patient admission to the hospital. Additionally, pharmacist involvement significantly improved appropriate selection and dosing of empiric antimicrobial therapy, based on respiratory culture history, for acute exacerbations within 24 h of admission (pre-pharmacist 48% vs. post-pharmacist 86.2% *p* ≤ 0.0001) and changing to appropriate definitive therapy within 24 h of available culture and susceptibility data (pre-pharmacist 70.2% vs. post-pharmacist 86.2% *p* = 0.024) [18]. 

Outpatient services include management of both chronic and acute medications and facilitation of direct admissions due to acute exacerbations. The clinical pharmacy specialist reviews each patient scheduled for clinic to assess medication regimens, culture and sensitivity results, and for adherence to guideline directed therapy. During a typical patient care visit, the clinical pharmacy specialist makes several interventions including initiating/modifying/discontinuing both chronic and acute medication(s), addressing adherence concerns, ordering labs for monitoring medication(s), and providing medication education. The clinical pharmacy specialist prioritizes patients that are acutely ill in clinic as well as ensures each patient sees the pharmacist at least annually. Due to improved efficiency in clinic workflow, the clinical pharmacy specialist is able to see most, if not all, patients scheduled in a given clinic day. Furthermore, the clinical pharmacy specialist is available for consultative services outside of the outpatient clinic schedule for other pediatric pulmonary clinics (e.g., nurse practitioner general pulmonary clinic). Inpatient clinical pharmacy services include facilitation of admissions in managing acute medications (e.g., antimicrobials) and necessary monitoring (e.g., laboratory tests, pharmacokinetic assessments), discharge medication consults, and patient/caregiver education. 

The clinical pharmacy specialist also collaborates with various members of the interprofessional care team. In improving medication adherence, the pharmacy collaborates with the psychologist and social worker, developing approaches to empower patients and encourage self-management. To improve medication adherence, the pharmacist, psychologist, and social worker may collaborate to formulate a strategic approach to facilitate adherence. Some strategies include, but are not limited to, dosing calendars, utilization of mobile applications, and family discussions on what is feasible with home schedules. The respiratory therapist and clinical pharmacy specialist work together to optimize appropriate use of inhaled medications including device technique. In partnering with the nutritionist/dietitian, medication–food interactions as well as safe and effective dosing of supplements including pancreatic enzymes and vitamins are optimized. Coordination of follow up items, such as laboratory testing, are improved with a good working relationship between the clinic nursing staff and our clinical pharmacy specialist. 

### 3.2. Growth of the Pharmacy Care Model for CSHCN-CMC

In 2016, the Cystic Fibrosis Foundation issued its first set of competitive grants to support expansion of outpatient pharmacy services including the addition of a certified pharmacy technician (CPhT) [19]. We added a CPhT in our practice model in 2016 to help improve medication access through facilitation of payer issues such as prior authorizations and connections to patient assistance programs. The CPhT’s role in our clinics has expanded beyond medication access support to include part of our pre-clinic visit planning processes. Approximately one week prior to a given clinic appointment, each patient/family scheduled for a given clinic date (e.g., 12–16 patients each full clinic date) receives a pre-clinic visit phone call from our CPhT. In this phone encounter, the CPhT conducts a preliminary medication history, provides reminders of items for the clinic visit (e.g., laboratory tests that have been ordered and need completion, “medication homework” from the pharmacist such as dosing logs), and collects patient/family questions or concerns for each discipline (i.e., respiratory therapy, pharmacy, social work, behavioral health, nutritionist, nurse, and physician). The information collected during the pre-clinic visit is then disseminated to the care team in preparing for each clinic. The CPhT is also actively involved in care team quality improvement initiatives such as improving patient/family clinic experience and coproduction of care. The CPhT has served as an effective partner for the clinical pharmacy specialist in facilitating complex prior authorizations and medication access issues as well as provide support in transition of care initiatives. The CPhT helps support the clinical pharmacist specialist in discharge medication planning and post-hospitalization logistics. The CPhT is responsible for initiating prior authorizations before discharge to eliminate patient barriers to medication access, ultimately ensuring patients are able to obtain prescriptions upon discharge. Furthermore, the CPhT contributes to post-hospitalization and clinic visit follow up related to medication use and monitoring by reaching out to patients to identify issues early to facilitate resolution. Given the complexity and challenges associated with medication use in this patient population, the duality of clinical pharmacy specialist and CPhT provide much needed medication-related expertise in the care of CSHCN-CMC.

From the quality improvement initiative evaluation of incorporating the CPhT into clinic workflow, there have been significant changes in the distribution of the clinical pharmacy specialist’s time, patient satisfaction, and other health care team satisfaction. For each given patient/family a maximum of three attempts are made by the CPhT to complete the pre-clinic visit call, with approximately 80% success. The CPhT impact include approximately $200,000 over a 12-month period in patient savings by connecting patients with assistance programs. The CPhT has successfully completed nearly 300 prior authorizations and approvals on average in a year, ultimately improving patient’s access to medication therapy. In evaluating the addition of a CPhT in our practice model, we found that the clinical pharmacy specialist has been able to increase frequency per clinic and overall time in providing medication education, adherence support, and administrative responsibilities (e.g., protocol development, research) (*p* < 0.001) [20]. 

## 4. Developing Future Interprofessional MCH Leaders 

The PPC traineeship curriculum provides various opportunities to learn about and participate in the care for underserved and diverse patient populations through the interprofessional didactic and practicum coursework, interprofessional clinical experiences, and interprofessional research opportunities. 

### 4.1. Didactic Learning and Project Collaboration Opportunities in the PPC Training Program

The traineeship begins with self-assessments and online trainee toolkits to orient learners to the MCHB and PPC to provide the trainee foundational knowledge regarding the interprofessional care of CSHCN-CMC. With a foundational knowledge, trainees progress into participating in interprofessional clinical experiences, didactic lectures, leadership projects, research, and self-evaluations/assessments to allow learners to develop and exercise necessary skills in providing family-centered, culturally sensitive care for patients with chronic respiratory conditions (e.g., cystic fibrosis, severe asthma).

The PPC core curriculum facilitates and encourages interprofessional collaboration for research and advocacy projects with the guidance and support of expert faculty. These interprofessional collaborative efforts between trainees have resulted in various initiatives improving community health through the development of community education programs and tools thereby by increasing patient and family access to resources and education. Additionally, trainee projects have engaged those in the community, such as school staff (e.g., teachers, school nurses, coaches), to educate and provide access to resources about the needs of CSHCN-CMC. Examples of PPC trainee and faculty interprofessional efforts to support patient/family and community engagement include CF patient/family education events and school health care training events. Furthermore, trainees collaborate to formally present all-encompassing, critical assessments of patient cases covering various aspects of care, including but not limited to, medical history and management, social concerns, growth and development, as well as access to resources. The presentations are delivered and evaluated by the PPC faculty, associated health care providers, and other trainees. PPC trainees also have the opportunity to help support and collaborate with various state Title V (MCH) agencies or other MCH-related programs, contributing to various types of service, training, continuing education, technical assistance, product development and research.

Trainees’ participation in didactic lectures and assigned readings spanning the traineeship year provides insight into various topics focusing on caring for children with chronic pulmonary disorders, encompassing specific issues related to each specialty discipline (Table 1). Topics include disease state reviews, approaches to community assessment, discipline roles in the care of CSHCN-CMC (through the lens of pulmonary disease such as CF and asthma), cultural sensitivity, palliative care, and considerations in MCH project/research development. Through the didactic curriculum, exposure to diverse education and training backgrounds allows trainees to gain respect and understanding of the role of the interprofessional team members and how to optimize collaboration. The didactic portion of the traineeship provides an all-embracing understanding to optimize a comprehensive patient care approach to CSHCN.

Beyond site-specific education and collaboration, the PPC traineeship facilitates collaboration between trainees from across the country to learn about family-centered care and gain insight into how various approaches are implemented in each institution. Online webinars and in-person meetings at the PPC Annual Meeting facilitate interaction and collaboration between trainees from all six national PPC centers. Real-time webinar sessions encourage open discussion about strengths and challenges in the provision of care for CSHCN-CMC and provide an additional modality for trainees to gain appreciation for geographic differences in practice, barriers, and health systems. As part of the annual PPC meeting, where all six centers come together to share current and develop new center-specific or joint initiatives, trainees have the opportunity to learn from, collaborate, and network with other centers’ PPC faculty and leadership in the MCHB.

Another significant component of the PPC traineeship is ensuring learners have the opportunity to gain interprofessional, first-hand clinical experiences to exercise and refine skills necessary to optimize patient care and outcomes. 

### 4.2. Interprofessional Clinical Training in the UA PPC

As the UA PPC faculty and care team collaborate in patient care and research, the trainees also function in a similar fashion. PPC trainees not only undergo leadership training together, but trainees also collaborate on various projects including case assessment and presentations, advocacy opportunities and community needs assessments, and they also work closely together in clinical care. The primary clinical setting for the traineeship is in outpatient, ambulatory care clinics (e.g., pediatric CF or severe asthma); however, exposure to acute care management and transition of care between inpatient and outpatient settings are included to emphasize the importance of a comprehensive care model. Each discipline contributes their expertise in the care of CSHCN-CMC, both faculty and trainee, with the common goal of providing quality, comprehensive care for CSHCN-CMC while developing future MCH clinicians and leaders. 

To unify the interprofessional team’s goals for each clinic day, the care team holds approximately 30-min pre-clinic briefing sessions to review each patient scheduled to be seen during that clinic date, typically 12–16 patients per clinic date. In these sessions, every patient case is reviewed as a team. Each discipline has the opportunity to discuss potential concerns, questions, and recommendations that must be addressed that day. During the pre-clinic briefing sessions, trainees and mentors participate in discussions as part of the comprehensive health care team to minimize duplication between disciplines and optimize the patient and providers’ time. Furthermore, the pre-clinic briefing sessions allow the care team to develop clear and unified message(s) for patients/families related to goals and coproduction of care.

At the end of each clinic, the interprofessional team again facilitates a 30-min post-clinic debriefing session. The debriefing session, similar to the pre-clinic briefing sessions, reviews each patient seen during the clinic to review the patient’s case. Discussion points include pertinent issues discovered during the visit, education provided to the patient and changes to therapy and necessary monitoring and/or follow up. This session closes the communication circle for a given clinic day to ensure all disciplines are in-tune with the goals established by the patient, family, and interprofessional care team members during the clinic visits.

These interprofessional clinical training opportunities allow trainees to progress through various stages of experiential learning (i.e., direct instruction, modeling, coaching, and facilitating), thereby having the opportunities to develop and refine critical skills for a future career in patient/family-centered, interprofessional care. In learning from the model set forth by PPC faculty of various disciplines, trainees learn to integrate themselves as a contributing member of the interprofessional team. 

### 4.3. The Layered Learning Model in the PPC Pharmacy Discipline Traineeship

While pediatric pharmacy practice has previously been associated with inpatient or acute care, the representation of pharmacists practicing in pediatric ambulatory clinics is currently limited. As such, there is opportunity to expand available experiences for pharmacy students and residents in a pediatric ambulatory care environment [21]. Since the addition of the pharmacy discipline to the UA PPC in 2009, the center has provided training and education for nearly twenty pharmacy trainees (i.e., pediatric pharmacy residents and pharmacy students) in the area of pediatric pulmonary ambulatory care. The pharmacy discipline traineeship in the UA PPC is based on an interprofessional, layered learning model. The layered learning model is described in the literature as having resident trainees overseeing other resident and/or student trainees. All of the trainees fall under the ultimate guidance and supervision of a more experienced clinical pharmacist [22]. Each year, one to two traineeships are offered at the student level, with the applicant pool recruited at the end of their first year of the Doctor of Pharmacy program at UA. Placement into the traineeship is a competitive process and the duration of the student traineeship is 24 months. The 24-month duration allows student trainees necessary time for professional degree responsibilities including internship. The addition of student trainees each year and continuation of trainees recruited from previous year, creates a senior student and junior student traineeship dynamic. This dynamic allows for peer-mentoring and collaboration opportunities. Additionally, each year, a post-graduate year 2 (PGY2) pediatric pharmacy resident(s) serves as a PPC resident trainee, with a duration of 12-months (i.e., their residency year). The resident serves as the most senior trainee of the PPC pharmacy discipline program and, in such a role, provides an added layer of mentorship and guidance to the student trainees. Residents have the opportunity to contribute to the education and training of student trainees as well as help mentor them on projects. The pharmacy faculty member (i.e., the clinical pharmacy specialist), resident, and student trainees meet regularly to discuss topics (e.g., disease states, journal club), pre-clinic round, and work on ongoing projects. 

In the clinical realm, the layer learning model with the pharmacy faculty member serving as a precepting pharmacist also provides opportunity for the resident to co-precept students in an experiential capacity. Both pharmacy student and resident trainees see patients, initially with all layers of learners present, and over time, the student seeing patients with the faculty or resident. Additionally, when appropriate, the pharmacy resident sees patients on their own, followed by discussion with the precepting pharmacist, physician, medical resident or fellow. Depending on clinical need and need for time efficiency, pharmacy trainees may also see patients with medical trainees, followed by discussion with precepting pharmacist and attending physician. All trainees document interventions, which are reviewed and cosigned by the precepting pharmacist. Trainees gain experience in the responsibilities of the clinical pharmacy specialist, including complex medication management which involves assessment of current therapy for adherence to clinical care guidelines and as appropriate, initializing, monitoring, modifying, and discontinuing medications, drug interaction management, adherence assessment and intervention, as well as patient/family education, as described above. Trainees learn about the role, logistics, and challenges of a pharmacy team (pharmacist and pharmacy technician) in the pediatric ambulatory care setting. Not only do they learn about the beneficial partnership between the clinical pharmacy specialist and pharmacy technician in this setting, but also the logistics of supporting and developing this pharmacy practice model. Given the partnership between pharmacist and physician at the UA PPC, trainees gain a strong appreciation for a collaborative practice model and its role in the comprehensive care of CSHCN-CMC.

### 4.4. Collaboration, Leadership, and Advocacy

In ensuring a comprehensive educational experience, trainees participate in a leadership training program as well as experience longitudinal mentorship from PPC faculty. Leadership development in the PPC traineeship is intended to bring awareness of and to optimize leadership qualities and styles. Learners are presented the opportunity to gain self-awareness of their personality and how it impacts their interactions with others. By exploring self-awareness and understanding, trainees may incorporate strategies when collaborating with others in leadership intensive workshops. Trainees have the opportunity to exercise leadership skills when learning how best to advocate for their patients and families. Advocacy, not only on a local level, but on a public policy standpoint, is a key component to providing quality family-centered, patient-centered, comprehensive care for CSHCN-CMC. Trainees learn to advocate for patients and caregivers through didactic experiences, interprofessional clinical experiences, and leadership training. Examples of activities related to community advocacy is through the participation in family-focused educational events, such as the Cystic Fibrosis Education Day. The educational event targets families and partners in caring for patients with cystic fibrosis to provide information necessary for all aspects of care (e.g., Social, medical, emotional). Not only are trainees learning to advocate for patients and families, trainees learn the importance of coproduction of care for patients with chronic pulmonary disorders. Trainees gain experience in advocacy on a public policy level through various modalities, such as preparing letters to members in Congress or Senate about a given issue and supporting Title V agencies through development of education programming or materials. 

Learning and exercising vital leadership, collaboration, and advocacy skills is critical for trainees as future professionals caring and advocating for CSHCN-CMC. The traineeship’s experiences help facilitate the development skills on how to impart positive change on a local level and beyond (e.g., larger organizational, community). Trainees will ultimately transition into becoming independent professionals with the skills to provide patient-centered, family-centered, comprehensive care for CSHCN-CMC. Through the opportunities afforded through the PPC training program, trainees are exposed to and practice the required skillset, knowledge, and characteristics necessary to establish themselves as a future MCH clinicians and leaders [23]. 

## 5. Future Needs and Directions

In the care of CSHCN-CMC, there exists a need to develop future clinicians and leaders, such as future pediatric pharmacists, in order to assure quality, comprehensive health care and excellent outcomes. PPCs are a prime example of an interprofessional training program that prepares future health care professionals to develop or improve community-based, family-centered health care for CSHCN-CMC, specifically, children with chronic respiratory diseases such as cystic fibrosis and asthma. Increasing awareness of such training opportunities and of its unique approaches to interprofessional practice are important to encourage similar collaborations in clinical education and practice at other institutions. The continuation of training programs, such as PPCs, is highly dependent on support from academic and health care institutions as well as extramural grant mechanisms, which may impart challenges on sustainability. Provider status and ability to bill for clinical and cognitive services (e.g., for pharmacists) may help provide a source of sustainability for clinical pharmacy services in pediatric ambulatory care settings. Though there is ongoing progress in the advocacy for pharmacist provider status, it is often associated with the adult ambulatory care realm and the need for advocacy of clinical pharmacy in the care of children, especially CSHCN-CMC, is needed. With the importance of safe and effective medication use in the care of CSHCN-CMC, clinical pharmacists should be a routine part of ambulatory care teams. At the UA PPC, the integration of a clinical pharmacy specialist has provided necessary expertise in safe and effective medication use and adherence as well as providing opportunities for future pharmacists (i.e., students and residents) to learn from and be part of an interprofessional care team in a pediatric ambulatory care setting.

## Figures and Tables

**Figure 1 children-06-00135-f001:**
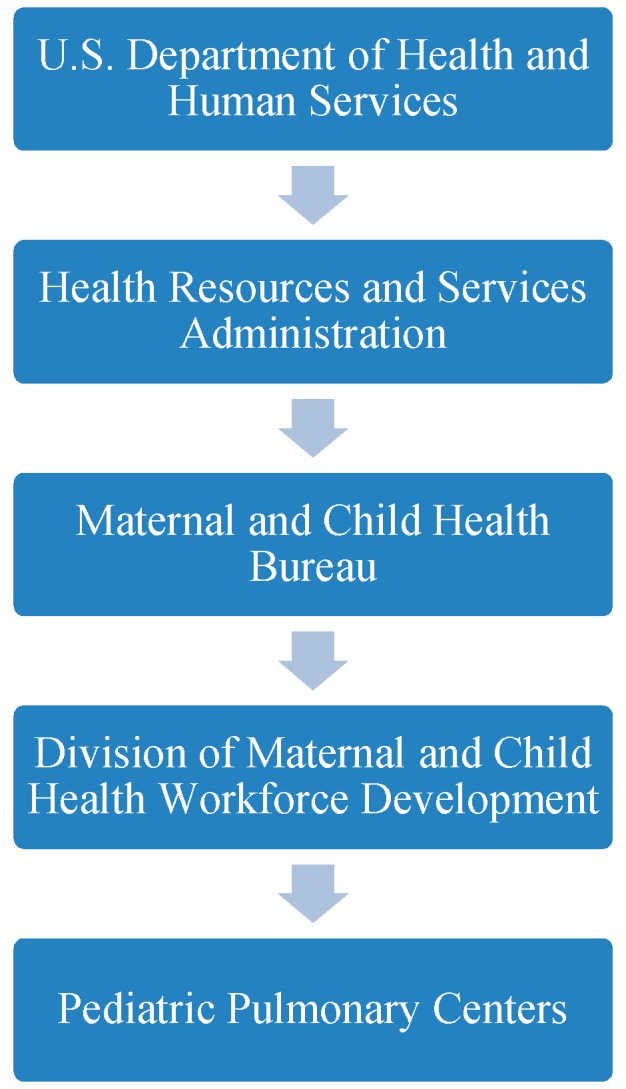
Organizational Oversight and Pediatric Pulmonary Centers (PPCs) [11].

**Figure 2 children-06-00135-f002:**
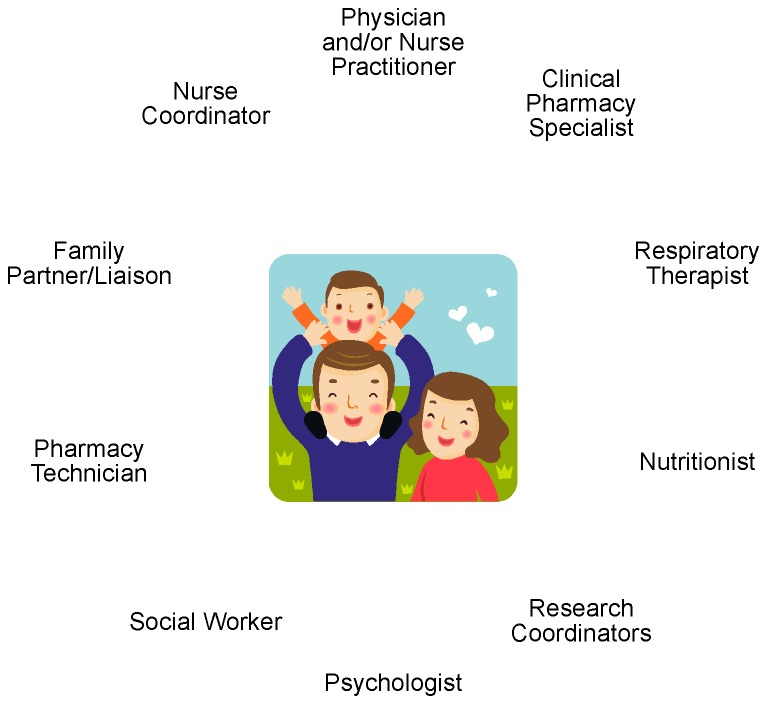
The University of Arizona Pediatric Pulmonary Center Interprofessional Patient Care Team Composition.

**Table 1 children-06-00135-t001:** PPC Example Curriculum Components.

**Lecture Series Topic**	**Faculty/Discipline Responsible for Teaching Topic**
Cystic Fibrosis and The Role of Newborn Screening	Pediatric Pulmonologist
Cystic Fibrosis Pharmacology	Clinical Pharmacy Specialist
Asthma Pharmacology	Clinical Pharmacy Specialist
Medication Adherence	Clinical Pharmacy Specialist
Research Ethics: An Introduction to the Ethical Conduct of Human Subjects Research	Pediatric Pulmonologist
Home Mechanical Ventilation	Pediatric Pulmonologist
Aerodigestive Disorders	Pediatric Pulmonologist
Working with Families of Terminally Ill Children	Registered Nurse
Caring for the Child with Diabetes	Registered Nurse or Certified Diabetes Educator
Introduction to Nutrition Care in Cystic Fibrosis Education Modules	Registered Dietician
Pulmonary Function Testing	Pediatric Pulmonary Fellow
Clinical Research Family Perspective	Family Leader
Role of Newborn Screening Family Perspective	Family Leader
Advocacy and Policy	Public Health Faculty or Professional
Transition to Adult Care	Social Worker
Child Life	Child Life Specialist
Community Needs Assessment	Public Health Guest Lecturer
**For Successful Completion of PPC Traineeship:**	
Online lecture series curriculum completion of associated assigned readings Life course theory evaluation Pre- and post-self-assessment Over 200 h of direct patient care in the clinical practice setting Attend and participate in the leadership workshop series Assessment and presentation of formal patient case(s) Development, completion, and presentation of capstone project	

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
