# Peer review of "Developing Future Clinical Pharmacy Leaders in the Interprofessional Care of Children with Special Health Care Needs and Medical Complexity (CSHCN-CMC) in a Pediatric Pulmonary Center"

_children, 2019, doi:10.3390/children6120135_

Round 1

Reviewer 1 Report

The authors satisfactorily addressed the reviewer's comments.  The paper is a unique contribution to the literature - in terms of developing the role of pharmacy for inter-professional practice.

Reviewer 2 Report

Thank you for the opportunity to re-review this manuscript, which outlines the importance of a pharmacist as part of an interdisciplinary health care team for children with medical complexities and the development of educational programs to train learners in the field.

The edits to the current version add to the overall readability and comprehension of the article.  Our earlier concerns have been adequately addressed. 

The objective data that are provided in the paragraph beginning line 243 are helpful in supporting the value of the pharmacy position.  However, the methods by which these data were collected and analyzed is not presented, and we would suggest doing so.
